# A simulation-based framework for modeling and prediction of personalized blood pressure trajectories in hypertensive patients after antihypertensive treatment

**Berit Hunsdieck**[1,2]*, **Johanna Mielke**[1☯], **Katja Ickstadt**[2,3☯], **Eren Elçi**[1☯]

**1** Bayer AG, Research & Early Development, Division Pharmaceuticals, Wuppertal, Germany,
**2** Department of Statistics, TU Dortmund University, Dortmund, Germany, **3** Lamarr-Institute for Machine Learning and Artificial Intelligence, Dortmund, Germany

☯ These authors contributed equally to this work.
\* berit.hunsdieck@bayer.com

**Data availability statement:** The underlying data used for further simulations is available via the UK biobank (see https://www.ukbiobank.ac.uk/) under Application #28807. The data yielding to the results presented in figure 4 (fig4.tif) are available via https://www.kaggle.com/datasets/berithu/logitudinal-blood-pressure.

**Funding:** The author(s) received no specific funding for this work.

## Abstract

Hypertension, a leading global cause of death, poses challenges in stabilizing blood pressure within target values despite various therapeutic options, often necessitating multiple therapy adjustments and delayed impact assessments. Recently, the first wrist-based wearable blood pressure measurement devices were introduced which allow for a continuous assessment of blood pressure trajectories. This enables the development of statistical methodology for prediction of saturated steady-state of blood pressure under treatment—and thus allowing physicians to adjust the therapy earlier. As a prerequisite for the evaluation of such models and algorithms, it is necessary to simulate reliable and realistic hypothetical patient trajectories under treatment with antihypertensive medication. In this paper, we propose a simulation framework for blood pressure profiles through Pharmacokinetic-Pharmacodynamic modeling, which incorporates individual daily rhythms, patient characteristics, and medication effects. We also propose and evaluate two models for steady-state prediction under antihypertensive therapy, a Gaussian process and a non-linear mixed effect model. When only one day of measurements is available, the Gaussian process is preferred, but in real-world situations with more data, the non-linear mixed effect model is favored. It effectively reduces RMSE and bias in noisy data, outperforming the Gaussian process regardless of sample size.

## Introduction

Hypertension is one of the leading causes of death worldwide. While there exist multiple therapeutic options, stabilization the blood pressure under recommended target values is challenging and often requires multiple adaptions of therapy. Traditionally, antihypertensive medicine is prescribed, and the impact of the therapy is only rechecked weeks after treatment

**Competing interests:** The authors have declared that no competing interests exist.

initiation. Globally, the number of hypertensive patients requiring anti-hypertensive medication increased from 594 million in 1975 to 1.3 billion in 2019, mainly in low- and middle-income countries (relative to growth in population) [1]. Starting at a blood pressure of 115/75 mmHg, the risk of death from a heart attack or stroke at ages 35 to 69 years doubles with every 20-point increase in systolic blood pressure, making it the leading cause of death [2]. High blood pressure accounts for nearly 10% of global health concerns [3]. According to the current state of the art, the efficacy of the anti-hypertensive drug is not checked in the doctor's office until a minimum of four weeks after it is taken for the first time. However, if the patient responds to the drug, an effect can be seen already after a few days [4]. Thus, an incorrect medication is identified only after weeks, although it could already be determined after a few days, whether the corresponding drug responds or not [4].

With the introduction of continuous blood pressure monitoring devices, early identification of ineffective medication becomes possible as this allows the physicians to continuously monitor (remotely) the impact of the therapy on blood pressure measurements. Physicians would be able to stop and adjust therapy early when it becomes clear that it is not leading to a steady-state blood pressure level (i.e., the state in which no additional change in blood pressure occurs as a result of further consistent administration) below the recommended targets. Current cuffless devices on the market are for example a chest patch developed by BioBeat Technologies [5], a wrist-wearable device developed by Aktiia [6], and a CAR-T ring developed by Skylabs [7]. We hypothesise that by making use of longitudinal data and statistical prediction models, sub-optimal blood pressure therapy can be detected even earlier by forecasting the expected saturated steady-state level of blood pressure as early as possible. This may help to identify patients with unsuccessful therapy even earlier.

Although none of these devices were accurately measuring blood pressure, they highlighted the need of re-evaluating the current practice of relying on a single blood pressure measurement in clinical settings [8]. Due to the novelty of wrist-based 24-hour blood pressure monitoring, there neither exists large data bases with relevant patient trajectories under recently initated blood pressure treatment nor simulation frameworks for generating artificial data. Therefore, for developing of forecasting methodology it is crucial to simulate realistic patient trajectories. In this paper, we propose such a simulation framework based on Pharmacokinetic-Pharmacodynamic (PKPD) modeling, which incorporates individual daily rhythms, patient characteristics, and medication effects. In order to integrate realistic patient covariate pattern, we integrate patient data from the UK Biobank [9].

Based on the simulated data, we develop two approaches to predict the steady state of a patient's blood pressure as early as possible. Our approach is, to the best of our knowledge, the first of its kind with a focus on antihypertensive therapy. Previous approaches aimed to forecast the blood pressure-lowering effect of antihypertensive medication on population level by identification of risk factors for unsuccessful drug titration for antihypertensive therapy. However, while these results are relevant for the overall population, none of them exhibits sufficient flexibility to ensure individual patient-dependent modeling. [10] introduced an initial approach to individual modeling based on general pharmacokinetic modeling. However, in this model the trajectory is predetermined by a fixed parametric form, rendering it less flexible. We optimize and tailor this approach to model patient trajectories under antihypertensive therapy. In addition, we explore Gaussian Processes (GP) which offer even more flexibility. The models are compared in a simulation study.

Based on simulated data incorporating the circadian rhythm, PKPD effects and measurements errors, the different models are evaluated. When only a single day of measurements is available, the Gaussian process is the preferred choice; however, in real-world scenarios with more data, the non-linear mixed effect model is preferred. It significantly lowers RMSE and

bias in noisy data, surpassing the performance of the Gaussian process regardless of sample size. Integrating the algorithm into wearable devices presents an innovative way to monitor medication adherence in hypertensive patients. It can predict the likelihood of achieving blood pressure reduction goals within the next day, allowing for timely adjustments to dosage and medication if necessary.

## A simulation-based framework for patient trajectories under antihypertensive therapy

The proposed framework aims to generate realistic patient data from continuous blood pressure devices under antihypertensive medication, i.e. a trajectory of the change of blood pressure values. We build our model on the following components:

1. Medication effect: The modeling of the medication effect is the core of our model; we use a PKPD model to ensure that the simulated data is realistic
2. Circadian rhythm: Since blood pressure varies during the day, we include the daily rhythm in the model.
3. Inclusion of uncertainty: Since in reality trajectories contain measurement errors, it is important to include within-day as well as day-to-day variability.
4. Realistic covariate patterns: We leverage UK Biobank data for getting realistic distribution of relevant covariates (e.g., age, sex) in the relevant patient population (hypertensive patient without pre-treatment)—it is known that these covariates influence the treatment effect and baseline blood pressure values.

Fig 1 provides an example of the individual components for one subject, illustrating the simulated effects on diastolic blood pressure, taking into account medication effects (a), circadian rhythms (b), and within-day variability. As different medication groups may yield to distinct patterns, the simulation will cover a large amount of variation, allowing for the prediction of diverse patterns independent of the underlying medication group. In the following, we will describe in detail the underlying mechanisms of this figure.

### PKPD medication effect

The PKPD modeling of the medication effect is the core of our framework. For each simulated patient, it is assumed that no medication is taken within the baseline period of two days (e.g. $t_{day} \in \{-2, -1\}$). The medication phase begins on day 0 ($t_{day} = 0$), with the medication being administered every morning at 8 a.m. from this point on. The corresponding timeline is illustrated in Fig 2.

The modeling of the medication effect is done by modeling the concentration-effect relationship. In pharmacology, this can usually be achieved by different PKPD modeling approaches. In a PKPD approach, the plasma concentration of the drug as a function of dose has to be modeled first by using a pharmacokinetic model. Given this time-dependent concentration, the effect can be modeled as a function of plasma concentration using a pharmacodynamic model. This scheme is illustrated in the Appendix, S1 Fig. In the following, we first describe the PK model and afterwards the PD model.

#### Pharmacokinetic modeling

Pharmacokinetic modeling uses compartment models to describe plasma concentration. These models vary in compartment number and absorption rates, such as single or three compartments in the literature. Although there are already complex compartment modeling approaches, a two-compartment model will be applied in the following analysis since

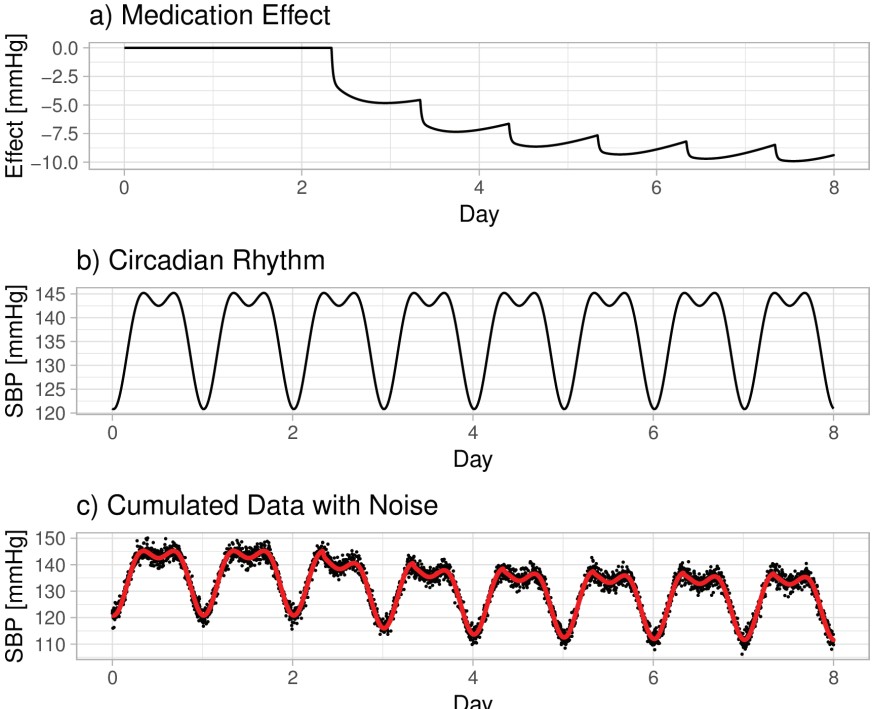

**Fig 1. Example: Simulated trajectory of diastolic blood pressure for a hypothetical patient over a four-day period.** (a) demonstrates the medication effect as predicted by a pharmacokinetic-pharmacodynamic (PKPD) model. (b) depicts the circadian rhythm's impact across the duration. (c) presents a combined view, with the circadian and medication effects overlaid in red, alongside the original data adjusted for within-day variability depicted in black.

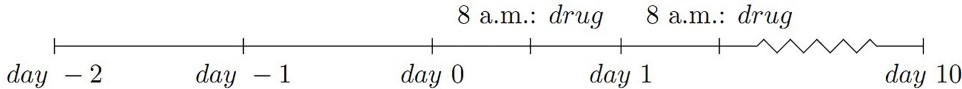

**Fig 2. Timeline of simulated drug administration.**

this model is mainly used for modeling medication effects of blood pressure lowering drugs [11–13] . This choice is justified by the fact that a two-compartment PKPD model offers a balanced combination of simplicity, appropriateness, and clinical relevance, making it a preferred choice in many applications [14]. We opt for a widely-used two-compartment effect model, outlined in [15] and [16], with central plasma, gut, and peripheral tissue compartments, as visualized in Fig 3.

The underlying equations for the compartment model illustrated in Fig 3 are given by

$$\frac{dC(1)(t)}{dt} = -K_a \cdot C(1)(t)$$
$$\frac{dC(2)(t)}{dt} = \frac{-(K_{10} + K_{21}) \cdot C(2)(t) + K_a \cdot C(1)(t) + K_{12} \cdot C(3)(t)}{VC}$$
$$\frac{dC(3)(t)}{dt} = K_{21} \cdot C(2)(t) - K_{12} \cdot C(3)(t) \quad . \tag{1}$$

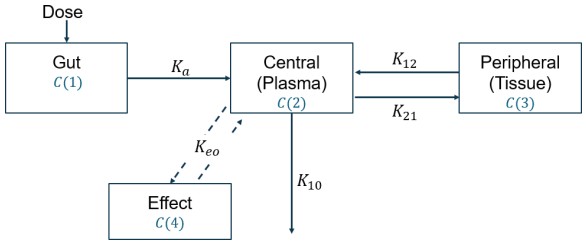

**Fig 3. Relationships within the pharmacokinetic model: Compartmental model parameters $C(1)$, $C(2)$, and $C(3)$ represent compartments (Gut, Central, Peripheral), $K_a$, $K_{10}$, $K_{21}$, and $K_{12}$ denote absorption and transport rates; Relationships within the Pharmacodynamic model: $C(4)$ represents effect compartment.** $K_{e0}$ denotes rate constant for transfer to the effect compartment.

The parameters are defined as

$$
\begin{aligned}
C(1) &: \text{Gut} \\
C(2) &: \text{Central (Plasma)} \\
C(3) &: \text{Peripheral (Tissue)} \\
K_a, \ K_{10} &:= \text{Rate of absorption} \\
K_{21} &: \text{Rate of transport of the drug from the central to the peripheral} \\
&\quad \text{compartment} \\
K_{12} &: \text{Rate of transport of the drug from the peripheral to the central} \\
&\quad \text{compartment} \\
K_{e0} &: \text{Rate constant for transfer to effect compartment.}
\end{aligned}
\tag{2}
$$

One important additional parameter is the volume of distribution (or volume capacity, *VC*), which is the volume of the central compartment. It relates the total amount of drug in the body to the plasma concentration of the drug at a given time. The following equation applies:

$$
VC[L] = \frac{\text{Amount of drug in the body [mg]}}{\text{Plasma concentration of drug [mg/L]}}
\tag{3}
$$

A drug with a high *VC* tends to leave the plasma and enter the extravascular compartments of the body. Conversely, a drug with a low *VC* tends to remain in the plasma, meaning that a lower dose of a drug is required to reach a given plasma concentration [17].

An increase of $K_{10}$ corresponds to the situation that the concentration cannot be held at a high level for a long time. In general, the concentration is lower and the steady state is reached faster. An increase in $K_{21}$ behaves similarly to an increase in $K_{10}$. The general plasma concentration reaches higher values and there are more significant peaks within one day. In addition, the steady state is reached more quickly. In contrast, an increase in $K_{12}$ leads to a longer time to reach the steady state, and the decline within individual days is weaker. The steady state is defined as the plateau where the effect will not increase further. An increase in $K_{e0}$ results in a slight increase in the effective height. An increase in $K_a$ affects the pattern of plasma concentration within a day. It leads to a steeper decrease.

Parameter choice involves considering factors like time to peak concentration (for blood pressure medication: 0.25 to 2 hours [18]) and how age and weight impact clearance and volume capacity [19,20]. A balance between literature values and desired characteristics is essential to avoid unrealistic drug effect curves. The selected corresponding parameters are listed in section "Evaluation of proposed prediction approaches".

**Pharmacodynamic modeling**

Given the plasma concentration the drug effect can be modeled by common pharmacodynamic models. The specific choice of the concentration-effect relationships needs to consider that, in real life, the effect of anti-hypertensives will reach the steady state after a few days. For this scenario, the model typically used in literature is the sigmoid $E_{max}$ (cf. [21], [22]).

Given the sigmoid $E_{max}$-model, the blood pressure lowering effect $BP_{eff}$ is defined as

$$BP_{eff} = E_{max} \cdot \frac{CE}{CE + EC_{50}} \quad , \tag{4}$$

where $CE$ is the drug concentration and $E_{max}$ can be interpreted as the maximum effect of a specific drug and $EC_{50}$ as the half-life time, e.g. the time period after which half of the effect has been achieved. These parameters vary depending on the drug and the subject.

In the literature, it is reported that the anti-hypertensive steady state effect is reached in approximately five to seven days based on calcium channel blockers [23,24]). This is used as a reference for simulation. The maximal effect ($E_{max}$) directly influences a drug's maximal effect, with baseline blood pressure and ethnicity playing roles. Higher baseline blood pressure leads to higher expected absolute drug effects [21]. Ethnicity can also impact drug effectiveness; for instance, the Angiotensin II receptor antagonist Eprosartan has little effect on blood pressure in Black Africans [25]. The selected corresponding parameters are listed in section "Evaluation of proposed prediction approaches".

**Circadian rhythm.**   Often the time of day is not taken into account when blood pressure is assessed [26]. However, for continuous blood pressure trajectories, this is considered a crucial information. We assume a circadian rhythm during the day, which is influenced e.g. by sleeping phases [27]. An example of such a pattern is displayed in Fig 4.

For the simulation, we use a simplified Fourier construction with two cosine functions [28], which yields

$$SBP(t) = BSL + amp_1 \cdot cos\left(\frac{2\pi \cdot (t + hor)}{24}\right) + amp_2 \cdot cos\left(\frac{2\pi \cdot (t + hor)}{12}\right) \tag{5}$$

with

$$nadir = \left(BSL - \frac{2}{3} \cdot change\right) \cdot (1 + exp(v))$$

$$amp_2 = \frac{1}{3} \cdot (BSL - nadir) - \frac{4}{9} \cdot change$$

$$\qquad - \frac{2}{9} \cdot \sqrt{6 \cdot change \cdot (nadir - BSL) + 4 \cdot change^2}$$

$$amp_1 = BSL - nadir + amp_2 \quad . \tag{6}$$

$SBP$ denotes the systolic blood pressure, $BSL$ denotes the baseline (systolic) blood pressure. The parameters *nadir* and *change* represent patient characteristics. First, *nadir* is defined as

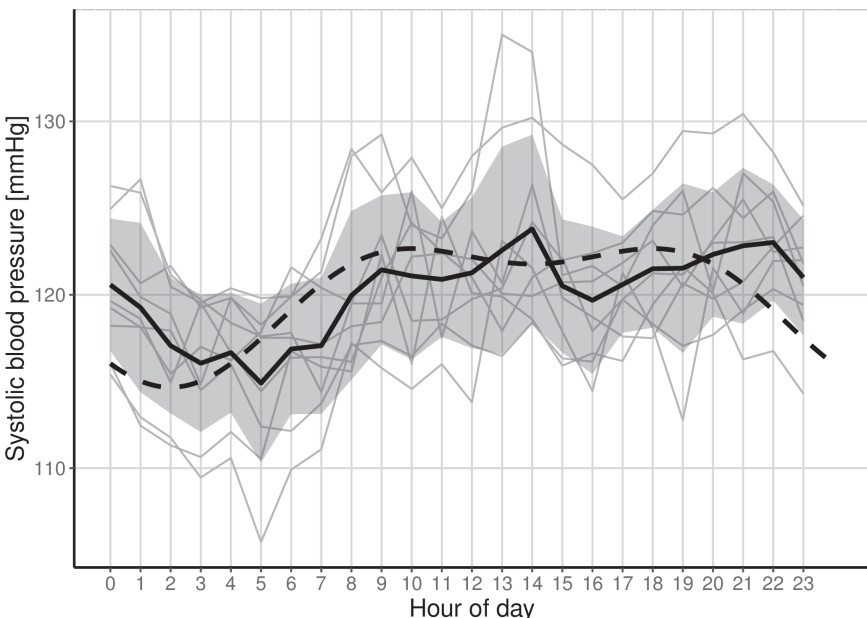

**Fig 4. Internal study data with eleven voluntary, healthy employees wearing a wearable device for a time frame of around two weeks.** In grey: Individual aggregated circadian rhythms; In black: Mean individual circadian rhythm; In black (dashed): Simulated circadian rhythm.

the minimum systolic blood pressure during the night. Second, *change* is defined as the difference between the maximum systolic blood pressure during the day and the minimum systolic blood pressure during the night. The corresponding parameters can be derived from data given by an internal study. In the internal study, eleven voluntary, healthy employees tested a wearable device for a time frame of around two weeks.

The individual trajectories as well as the derived simulated trajectory based on internal study data are given in Fig 4. The specific parameter selection, which results from this, can be found in the subsequent section.

**Variations from the ideal curve: Within-day and day-to-day variability.** Noise on the data should be added for capturing the measurement error present in real-world data. This noise can fluctuate within a day (intra-daily, within-day) as well as between different days (inter-daily, day-to-day noise).

To account for additional noise within a day, such as coffee consumption, individual lifestyle, or environmental impacts, we can introduce variability by adding random noise to each measurement. For this intra-daily variability, we rely on data from current marketed wearable devices, such as Aktiia (see https://aktiia.com/de/, accessed on May 11, 2024). It was estimated that the within-day variability of individual measurements is approximately 5 mmHg. This is consistent with the variability measured in an internal study as well the variability given in [29].

We assume that the intra-daily variability is independently normally distributed around zero. When assuming that 5 mmHg corresponds to the 95%-quantile of this distribution, the final distribution is given by

$$noise_{intra} \sim \mathcal{N}\left(0, \left(5/z_{1.99/2}\right)^2\right) = \mathcal{N}\left(0, 1.94^2\right)$$

for the intra-daily variability.

In addition, we assume inter-daily variability given by about $\pm 8$ mmHg. This can be validated by the internal study as well. Inter-daily variability in the measurements can be caused by unusual activities like doing sports or being sick. As before, assuming that the inter-daily variability is normally distributed around 0, the corresponding standard deviation can be derived by assuming 8 mmHg as the 99% quantile of it's distribution, which then yields to

$$noise_{inter} \sim \mathcal{N}(0, (8/z_{1.99/2})^2) = \mathcal{N}(0, 3.11^2) \quad .$$

## Covariate-specific impact on blood pressure trajectories

We have already mentioned that covariates like the body weight can have a huge impact on the blood pressure as well as on the blood pressure lowering effect of anti-hypertensives [20] and, therefore, some of the parameters introduced above for the PKPD-model are dependent on patient specific covariates. A summary of the hypothesized relationships between (clinical) covariates and the individual submodels is presented in Fig 5: e.g, we assume that weight, age and sex both influences the $VC$ and baseline blood pressure. Ethnicity influences the maximum effect of the antihypertensive drug. Indirectly, all components contribute to the individual blood pressure profiles. The individual literature sources can be found in the Appendix, S2 table.

Assuming that we have simulated a dataset with realistic patient characteristics, we now need to simulate a blood pressure trajectory per patient. For that, we need to consider that it is known that some parameters from the PKPD model need to be adjusted to account, for example, for different sex and age. More concretely, according to Fig 5, the above introduced parameters from the PKPD model, $E_{max}$, $VC$, and $BSL$, need to be adjusted by modeling the impact of the patient-specific covariates (relative to the population mean) by

$$\mu_{BSL_i} = BSL_0 + \alpha_{age} \cdot (age_i - \overline{age}) + \alpha_{sex} \cdot sex_i + \alpha_{weight} \cdot (weight_i - \overline{weight})$$
$$\Rightarrow BSL_i \sim \mathcal{N}(\mu_{BSL,i}, sd_{BSL}^2)$$
$$\mu_{VC,i} = VC_0 + \gamma_{age} \cdot (age_i - \overline{age}) + \gamma_{weight} \cdot (weight_i - \overline{weight})$$
$$\Rightarrow VC_i \sim \mathcal{N}(\mu_{VC,i}, sd_{VC}^2)$$
$$\mu_{E_{max},i} = E_{max,0} + \beta_{BSL} \cdot (BSL_i - \overline{BSL}) + \beta_{ethn} \cdot (ethn_i - \overline{ethn})$$
$$\Rightarrow E_{max,i} \sim \mathcal{N}(\mu_{E_{max,i}}, sd_{Emax}^2) \tag{7}$$

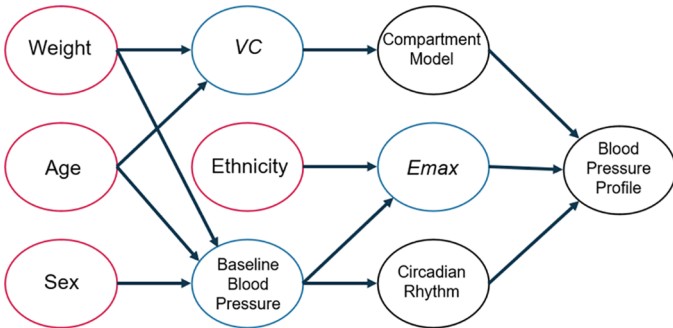

**Fig 5. Impact of covariates on simulated blood pressure profiles; red: Clinical covariates, blue: Variables affected by covariates essential for PKPD modeling and circadian rhythm, black: Models encompassing circadian rhythm and medication effects.**

The above introduced parameters from the PKPD model, $BSL_0$, $VC_0$, and $E_{max,0}$, are the baseline parameters without any influence of additional covariates. The factors $\alpha_{age}$, $\alpha_{sex}$, $\alpha_{weight}$, $\gamma_{age}$, $\gamma_{weight}$, $\beta_{BSL}$, and $\beta_{ethn}$, are derived by literature review (e.g., see [21,30,31]) as well as explorative analyses so that the effects given in literature are well expressed. The final values are given in the Appendix, S5 table. The corresponding reference values, i.e. for ethnicity, are given by the population means, i.e. denoted by $\overline{ethn}$. For the categorical variable ethnicity, $\overline{ethn}$ = 0 denotes people of white ethnicity.

## Models for forecasting individual patient trajectories

Based on the simulated data, the aim is to model the drug effect on blood pressure and predict the effect course of anti-hypertensive medication in the early stages of titration phases. Since the individual courses within a day are irrelevant for this goal, the daily average values are considered. In this section, we develop a non-parametric (GP) and a parametric (nlme) which can be used for forecasting individual patient trajectories. Let the time courses be aggregated by the daily means of the systolic blood pressure for $N$ individuals given in the present data set. In addition, the covariates height, weight, age, sex, and ethnicity are assumed to be known for all individuals.

In the following, $i$, $i \in \{1, \dots, N\}$ denotes the individual patient ID and $t$, $t \in \{0, \dots, n_o\}$ the day of observation. $t_{max}$, $t_{max} \in \{1, \dots, n_m\}$ is the day of last known observation in test data, which is constant over all test individuals. Further, $y_i(t)$ is the blood pressure value of individual $i$ at day $t$. The corresponding predicted values of individual $i$ at day $t$, $t \in \{1, \dots, n_o\}$, are denoted as $\hat{y}_{i,t_{max}}^{(GP)}(t)$ and $\hat{y}_{i,t_{max}}^{(nlme)}(t)$ based on the GP model and nlme model given $t_{max}$ data points, respectively.

### Parametric approach: Non-linear mixed effects model

In cases where additional information concerning the shape of a curve is available such information can be harnessed by a parametric model to optimize the parameters of interest. A widely adopted model for this purpose is the mixed effects model, which enables the estimation of both population-wide influences and subject-specific effects. In our use case, the model formula is motivated by [10], where the blood pressure reduction $BP_{Reduction}(t)$ at time $t$ is modeled by an asymptotic approach to a plateau given by

$$BP_{Reduction}(t) = R_{max} \cdot \left( 1 - exp\left( -\frac{t \cdot ln(2)}{dt_{1/2}} \right) \right) \tag{8}$$

with $R_{max}$ as the maximal reduction in blood pressure, $dt_{1/2}$ as dynamic half-life time, i.e., time to reach 50% of maximal reduction, and $t$ is the time since the beginning of therapy or taking medication.

To account for uncertainty in the analysis and enable more robust inference and predictive accuracy despite limited data, we will adopt a Bayesian modeling framework.

For our use case, we need to adjust the model given by formula (8) as our objective is to model individual patient trajectories. This specifically means that we need to allow for patient-specific saturation levels $R_{max}$ and patients-specific half lives $dt_{1/2}$. We also introduce an additional population-wide parameter $\omega$, that allows for additional variations in formula (8). To avoid high complexity and computational intensity of the model, an individual effect on $\omega$ is not considered, as it exerts no direct influence on effect magnitude or time to full effect. In other words, we assume that the general curve shape depends on $\omega$, with variations

limited to half-life time and effect magnitude among individuals. To ensure that the estimators remain in the positive range, we log-transform the parameters, resulting in the following equation for the likelihood:

$$\hat{y}_{i,t_{max}}^{(nlme)}(t)|_{\eta_i} = exp(lR_{max,i,t_{max}}) \cdot \left( 1 - exp\left( -\left( \frac{t \cdot ln(2)}{exp(lt_{1/2,i,t_{max}})} \right)^{exp(l\omega)} \right) \right) \tag{9}$$

$$lR_{max,i,t_{max}} = lR_{max,pop} + \eta_{i,1,t_{max}}, \quad lt_{1/2,i,t_{max}} = lt_{1/2,pop} + \eta_{i,2,t_{max}}$$

with $\eta_{i,t_{max}} = (\eta_{i,1}, \eta_{i,2})^T$. The priors for the Bayesian framework are given by

$$lR_{max,pop} \sim \mathcal{N}(1.2, 0.1), \quad lt_{1/2,pop} \sim \mathcal{N}(0.5, 0.2), \quad l\omega \sim \mathcal{N}(1, 0.1) \quad . \tag{10}$$

resulting from analysing the shape with regard to the known properties regarding mean half-life time and mean effect size. The model is fitted using Bayesian techniques using the *brms* R package [32]. For this model, we assume homoscedasticity, normal-distributed residuals and independence of individuals.

We assume that data is split in a training dataset (which is used for fitting the model) and then applied to unseen observations. In practice, the first step needs to be completed prior to the first application of this approach whereas the application to unseen observations represents the realistic scenario for a use in practice (novel patients need to get a forecast).

## Non-parametric approach: Advanced Gaussian process

GPs provide a non-parametric method for estimating functions, allowing for the fitting of flexible models which capture the data while quantifying uncertainties in predictions. In hierarchical GPs, the data follows a nested structure given by a population-based GP and an individual GP. The population-based GP captures broad trends, while the individual GPs link observations to these trends. Prior knowledge enhances predictions [33,34].

To address challenges with convergence of Bayesian estimation as well as extrapolation problems using GPs, we propose an alternative approach: The proposed step-wise procedure, here called advanced GP, combines classical likelihood estimation methods and Bayesian methods in GP modeling. The analytical process consists of fitting a mean population GP to capture global trends. Residuals are then calculated. A Bayesian approach refines the model by estimating individual effects using training data. This completes the model building. For an estimation of effects on previously unseen data, we match the unseen observations to the subjects from the training data, identify trajectory-like data and make use of those for an accurate forecasting of the blood pressure saturation levels. More concretely, we follow these steps:

1. *Fit of Mean Population Gaussian Process:* In the initial step of our analysis, we fit a mean GP $\hat{y}_{mean,GP}(t)$ given by

$$\hat{y}_{mean,GP}(t)|(y_i(t), t) \sim \mathcal{N}\left( \mu_{mean}(t), \Sigma_{mean}(t) \right) \tag{11}$$

given the squared exponential kernel

$$k(x, x_\star) = \sigma^2 \cdot exp\left( -\frac{(x - x_\star)^2}{2l^2} \right) \quad . \tag{12}$$

Therefore, first the hyperparameters $\boldsymbol{\theta} = (l, \sigma)$ are fitted by optimizing the log marginal likelihood

$$ln\, p(\boldsymbol{y}|\boldsymbol{X}, \boldsymbol{\theta}) = -\frac{1}{2}\boldsymbol{y}^T\boldsymbol{K}^{-1}\boldsymbol{y} - \frac{1}{2}ln\,|\boldsymbol{K} + \sigma_n\boldsymbol{I}| - \frac{n}{2}ln(2\pi) \quad , \tag{13}$$

given the training data $\{(t, y_i(t)) : t \in \{0, \dots, n_o\}, \text{individual } i \text{ included in training subset}\}$ and $\boldsymbol{K} = (k(t_i, t_j))_{i,j, \in \{0, \dots, n_o\}}$. To achieve this, we employ the GPy Python package [35], which provides the necessary tools and functionalities for GP modeling. Given the fitted kernel $\hat{k}$ for the fitted hyperparameters $\hat{\boldsymbol{\theta}} = (\hat{l}, \hat{\sigma})$, the fitted GP is used to estimate $f_\star$. $f_\star \in \mathbb{R}^{n_o+1}$ denotes the the estimated mean population values, which are be derived by

$$f_\star | (\boldsymbol{X}_\star, \boldsymbol{X}, f) \sim \mathcal{N}(\hat{\boldsymbol{K}}(\boldsymbol{X}_\star, \boldsymbol{X}) \cdot \hat{\boldsymbol{K}}(\boldsymbol{X}, \boldsymbol{X})^{-1}f,$$
$$\hat{\boldsymbol{K}}(\boldsymbol{X}_\star, \boldsymbol{X}_\star) - (\hat{\boldsymbol{K}}(\boldsymbol{X}_\star, \boldsymbol{X}) \cdot \hat{\boldsymbol{K}}(\boldsymbol{X}, \boldsymbol{X})^{-1}(\hat{\boldsymbol{K}}(\boldsymbol{X}, \boldsymbol{X}_\star)), \tag{14}$$

with $(\boldsymbol{X}, f) = \{(t, y_i(t)) : t \in \{0, \dots, n_o\}, \text{individual i included in training subset}\}$ given $\boldsymbol{X}_\star = (0, \dots, n_o)$.

2. *Calculation of Remaining Residuals:* After fitting the mean GP model, we calculate the remaining residuals

$$\epsilon_i(t) = y_i(t) - \hat{y}_{mean,GP}(t) \quad . \tag{15}$$

These residuals represent the differences between the observed data points and the values predicted by the GP model.

3. *Bayesian Fit of GP for Training Data:* Moving forward, we transition into a Bayesian framework for model fitting. Specifically, we apply Bayesian methods to fit GP models on the remaining residuals $\epsilon_i(t)$, so that the corresponding estimates $\hat{\epsilon}_i(t)$ are given by

$$\hat{\epsilon}_i(t) \sim \mathcal{N}\left(\mu_{indiv}(t), \Sigma_{indiv}(t)\right) \tag{16}$$

at the individual level within the training data set. This Bayesian approach allows us to quantify uncertainties, and make probabilistic predictions based on the observed data.

4. *Bayesian Gaussian Process Modeling for Test Data:* When dealing with the test dataset, we employ a slightly different strategy. We identify a pre-selected number of individuals from the training data that are closest to each test case. To determine this proximity, we use a distance measure, specifically the smallest Euclidean distance

$$dist_{t_{max}}(\epsilon_i, \epsilon_j) = \sqrt{\sum_{t=1}^{t_{max}}(\epsilon_i(t) - \epsilon_j(t))^2} \tag{17}$$

computed from the residuals of the GP models calculated in step 2. The data of these five nearest individuals are selected to inform the subsequent modeling step and to serve as the basis for our Bayesian fitting of a GP. The selection process is driven by the need to have a representative and informative subset for modeling the test data, allowing us to leverage the information contained within the training data set. The resulting distribution is given by

$$\hat{\epsilon}_i(t) | \boldsymbol{S}_{nearest, i, t_{max}} \sim \mathcal{N}\left(\mu_{indiv}(t), \boldsymbol{\Sigma}_{indiv}(t)\right) \quad , \tag{18}$$

where $S_{nearest,i,t_{max}}$ contains all data of the nearest training IDs and the data up to day $t_{max}$ of individual $i$.

Combining the results of step 1 to step 4, the individual effect estimate $\hat{y}_{i,t_{max}}^{(GP)}(t)$ for individual i at time t is given by

$$\hat{y}_{i,t_{max}}^{(GP)}(t) = \hat{y}_{mean,GP}(t) + \hat{\epsilon}_i(t) .  \tag{19}$$

## Evaluation of simulation framework and comparison of statistical models

In this section, we describe how we use the proposed simulation framework to generate realistic patient trajectories. Afterwards, we explain how we test the two proposed forecasting approaches, namely the non-linear mixed-effects model and the advanced GP. We aim to compare the performance of the parametric and the non-parametric model for different simulation settings.

### Simulation of realistic patient characteristics

In a simulation study, it is necessary to simulate a realistic set of covariates of each patient. In order to generate realistic patient-dependent datasets, we need to ensure that the underlying baseline characteristics are representative for a patient population (with high untreated blood pressure). Modeling realistic distributions of the covariates (e.g., generating a dataset with artificial patients with specific sex, age, height) can be achieved by mimicking a given data set of hypertensive patients and their covariates. One way to obtain a large data set of hypertensive patients without medication is given by the UK Biobank [9]. The UK Biobank gives access to detailed phenotype information as well as real world information regarding health status, genetics, clinical tests and many other parameters for around 500.000 adults across the UK. For the retrieval of realistic distributions of covariates, we focus on on patients reported not to take any antihypertensive medication, but still having a diastolic blood pressure of at least 90 mmHg or a systolic blood pressure of at least 140 mmHg which corresponds to so-called hypertension stage 2 (see Appendix, S1 table).

### Evaluation of proposed prediction approaches

We aim to compare the proposed approaches in a simulation study. For that, we conducted an analysis based on 50 bootstrap test data sets. We employed a train-test split ratio of 2/3 to 1/3 to evaluate the model's performance. For the patients from the training dataset, all observations from day 0 to day 10 are included. For patients from the test dataset, we aim to mimic the situation in practice and therefore only include data for the one individual of interest with data spanning from day 0 to day $t_{max}$ (where $1 \le t_{max} \le 5$). We aim to predict the blood pressure level at day $t = 10$.

We vary the number of days to be included for the test data ($t_{max}$) from 1 to 5 to identify the minimal number of observations for solid predictions. We vary the number of subjects ($N \in \{60, 120, 200\}$) to estimate a minimum sample size for the different approaches. Lastly, the influence of different numbers of neighbours for the GP are compared given two cases, namely 5 and 10 nearest neighbours.

The PKPD parameter choices for simulating the corresponding patient trajectories can be found in S3 table.

For the evaluation of the proposed methodlogy, we focus on two different strategies: Firstly, the model-based predicted blood pressure curve can be compared with the true "raw"

value (without noise). However, in a real-world setting, this value is always influenced by noise. Therefore, it is sensible to consider a second scenario, in which the estimated value is compared with the underlying measured noisy value, i.e. the observed value. We assess the models' quality of fit by calculating the bias and the root mean squared error (RMSE) per day.

## Results

As outlined in previous chapters, the covariates must be simulated first in order to simulate the trajectories. This will be done based on the UK Biobank. A normal distribution is assumed, whereby the mean value and the standard deviation are estimated based on the UK Biobank cohort. Therefore, the UK Biobank patients with a systolic blood pressure of at least 140 mmHg or a diastolic blood pressure of at least 90 mmHg and no ongoing medication are obtained. The distribution of age and weight, separated by sex and ethnicity, is given by Table 1.

The resulting simulated dataset is based on these simulated covariates. Three concrete examples of simulated patient trajectories are given in Fig 6. The simulation incorporates both individual covariate-specific effects as well as within-day and day-to-day variability. It is noted that if specific characteristics of a mediciation class would be known (e.g., a specific half life), these could be easily integrated into the simulation framework. First, we compare the red lines (without noise) and note that the individual patient characteristics (gender, age, height, size, ethnicity) highly influence the observed trajectories. For example, Patient 2 observes a higher reduction of blood pressure. The introduced variability leads to further variation which are visible in the observed data (black dots).

Before systematically comparing the properties of the proposed forecasting approaches in the described simulation study, we aim to illustrate the characteristics in a single-patient example. In Fig 7 an example of individual predictions based on the different model types are given. The cohort size is given by $N = 200$. The forecasting tasks changes based on the available information from left to right: In the figure on the left, blood pressure measurement has only been measured for a single day after initiation of treatment. From left to right, more and more days of observations are given so that on the right, we have five days of blood pressure measurements given, which can be used for predicting the further course of blood pressure. In yellow, the corresponding predicted curve of using the GP with 5 nearest neighbours is given. In blue the non-linear mixed effects model is used for prediction. The black curve marks the underlying real trajectory without noise. It has been observed that, with limited information, the GP estimates the true curve more accurately than the nlme model. However, as more days of measurements are provided, the nlme model begins to outperform the advanced GP.

**Table 1. Summary statistics of covariates age (in years), weight (in kg), and sex in patients with hypertension stage 2, i.e., diastolic blood pressure greater than 90 mmHg or systolic blood pressure greater than 140 mmHg (see Appendix, S1 table), in the UK Biobank data.**

| Sex | Ethnicity | Covariate | Mean | Standard deviation | Cohort size |
|---|---|---|---|---|---|
| Female | Black | Weight | 82.3 | 16.8 | 2.228 |
| | | Age | 54.2 | 7.94 | |
| Female | White | Weight | 73.3 | 14.7 | 109.327 |
| | | Age | 59 | 7.19 | |
| Male | Black | Weight | 86.5 | 14.4 | 1.025 |
| | | Age | 51.6 | 7.95 | |
| Male | White | Weight | 86.3 | 13.8 | 79.705 |
| | | Age | 57.2 | 7.92 | |

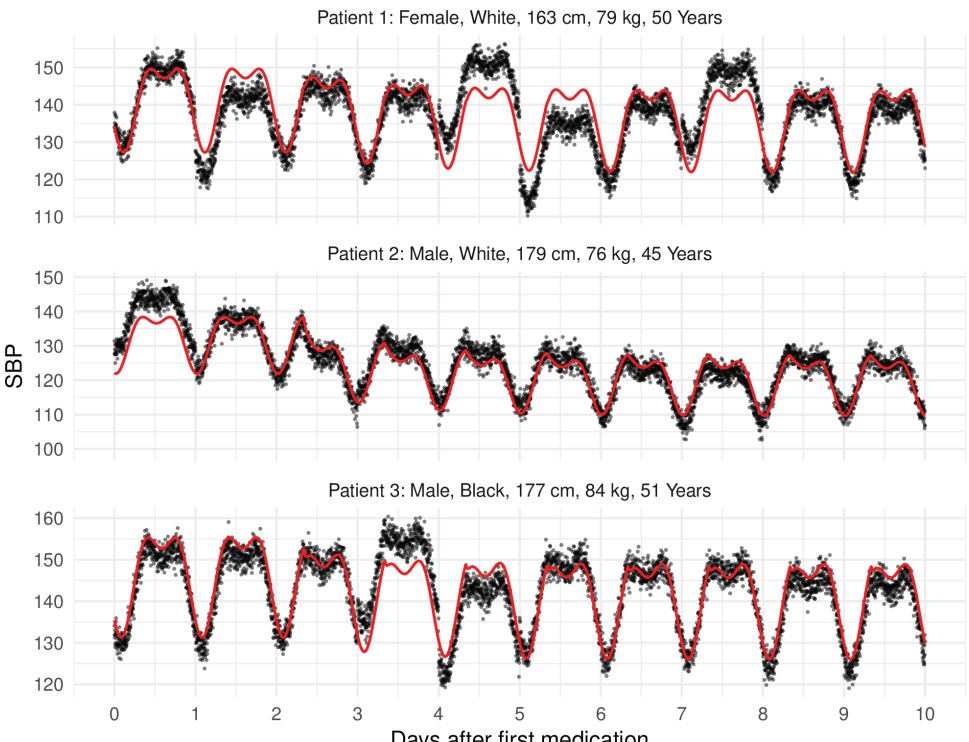

**Fig 6. Simulated trajectories of systolic blood pressure after taking anti-hypertensive medication of three hypothetical patients.** The red line corresponds to the raw blood pressure curve without any noise.

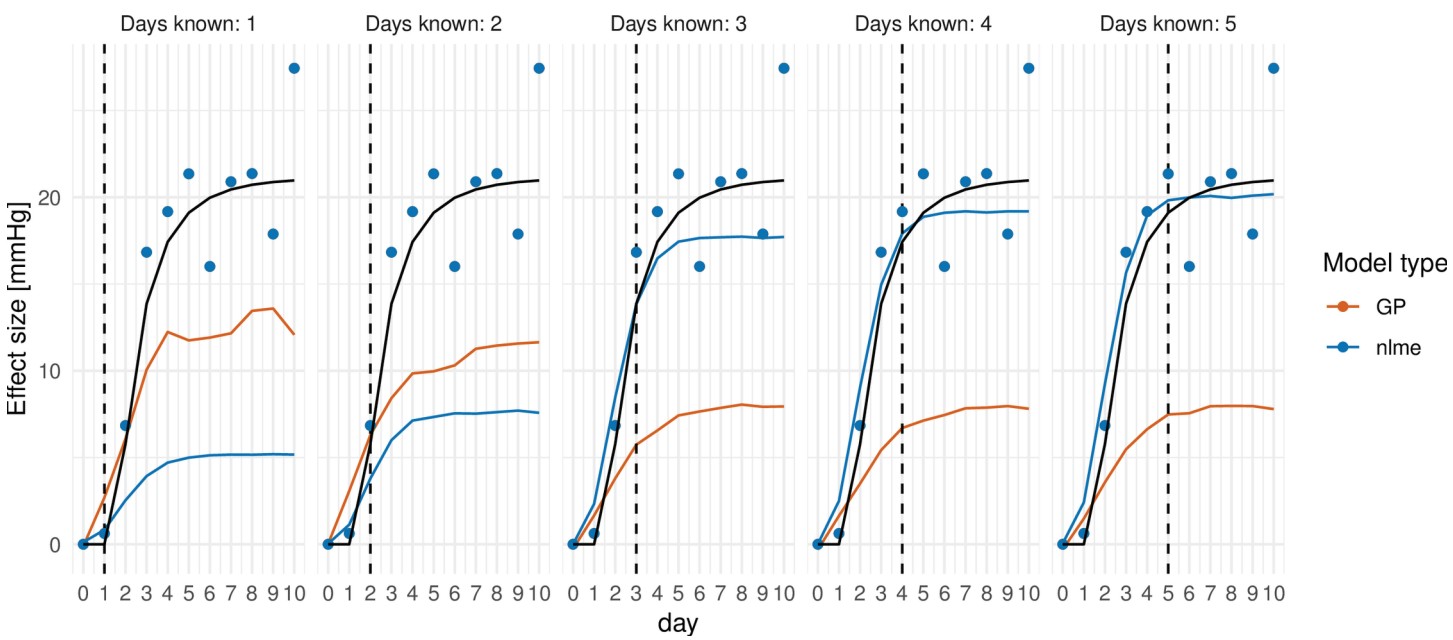

**Fig 7. Prediction of individual effect curve using a non-linear mixed-effects model (nlme) and an advanced Gaussian process model (GP) with five nearest neighbours, given $t_{max}$, $t_{max} \in \{1, \dots, 5\}$, days of measurements.** In black: true curve of medication effect. The dots represent the observed values.

Specifically, after five days of measurements, the estimated trajectory closely aligns with the true (unobserved) trajectory.

Generalising this to the full cohort with $N = 200$ patients, we will further focus on the comparing the predicted blood pressure values to the observed (noisy) values since in a real-world data setting, the true underlying values without any noise are not accessible. To ensure robustness, 50 datasets are simulated and the performance is compared.

First, the bias and rmse results are evaluated on a daily basis as illustrated in Fig 8. With a maximum absolute bias of approximately 1 mmHg all models are almost unbiased. While for the nlme model, we observe that adding in more information, i.e., increasing the number of available days with measurement, improves the performance by lowering the RMSE, the same is not true for the GP models: there, the RMSE is not reduced if more data is added. Looking at the RMSE and bias based on the noise-free values (see Appendix, S2 Fig), similar observations can be made.

One possible reason for this could be that the non-linear mixed-effects model implicitly has access to all training data during fitting. When NLME models are fitted to the data, they integrate information from both the fixed and random effects. The random effects are capturing the variation between different entities or subjects in the data set. In the GP models, only the population mean of these data points influences the model, while the individual fits of the residuals have access to only a fraction of the training data. Therefore, the NLME model

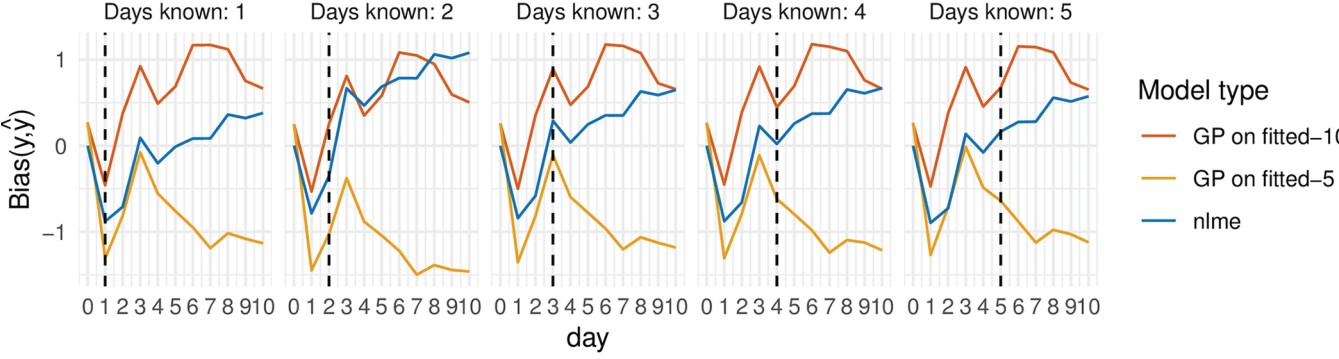

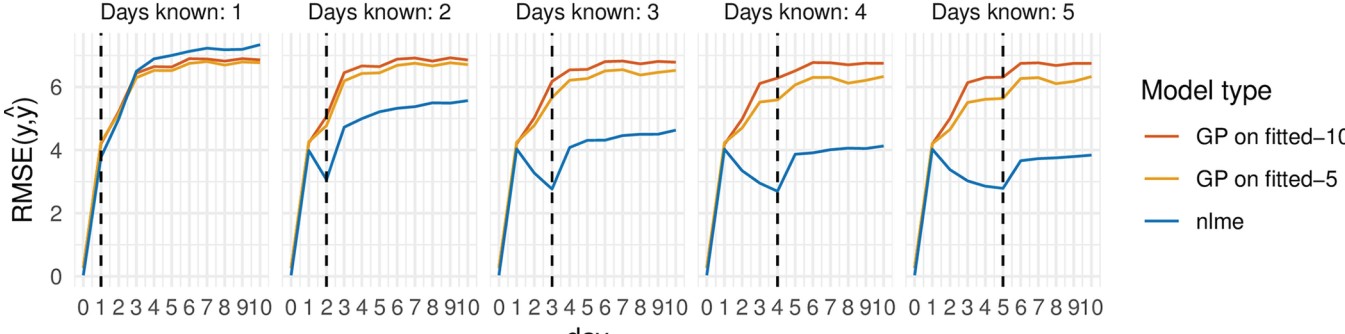

**Fig 8. Root mean squared error (RMSE) and bias across days** ($t = 0 \dots , 10$) **compared to actually observed values, models (non-linear mixed-effects model (nlme) in blue, Gaussian process (GP) with five resp. ten nearest neighbours in yellow resp. red)** ($t_{max} = 1, \dots , 5$, **vertical line**).

may have the capability to fit the data more accurately, leading to a lower RMSE. This could be especially true when the data contain a significant amount of noise or individual variability.

Looking at the performance of both models, the advanced GP performs slightly better for a single day in terms of RMSE, while the nlme approach demonstrates improved performance over time based on the bias as well as the RMSE. Therefore, the GP is the preferred method only if very limited data is available.

These findings appear also independent across a wide range of realistic sample sizes ($N$ = 60 to $N$ = 200). We also note that there is no major gain in performance if more subjects are added (see Appendix, S3 Fig) indicating that the saturation level in terms of performance is reached early.

## Conclusion and outlook

This paper presents a novel simulation framework for generating blood pressure profiles, which integrates Pharmacokinetic-Pharmacodynamic modeling with individual daily rhythms, patient characteristics, and medication effects. This model is very transparent and highly flexible and therefore can be adjusted to different medication classes. Also, the underlying patient characteristics are taken into account—allowing for tailoring to a target patient population, for example if inclusion/exclusion criteria for a clinical trial are already known.

Additionally, two models for predicting steady-state responses under anti-hypertensive therapy are developed and applied, namely an advanced Gaussian process and a non-linear mixed effect model. In scenarios where only one day of measurements is available, the Gaussian process is the preferred choice. However, in the real-world setting including more data is preferred. When multiple days of data are available, the nlme model becomes the preferred choice. It significantly reduces the RMSE as well as the bias for data with noise compared to the Gaussian process and demonstrates superior performance independent of the given sample size.

The algorithm's potential integration into wearable devices offers a innovative solution for monitoring medication adherence in hypertensive patients. The algorithm has the capability to provide a current probability or forecast of whether the patient will achieve the desired blood pressure reduction goal within the next day. This information can be used to make early adjustments to the dosage and medication if needed.

A growing interest has emerged in forecasting the individual effects of anti-hypertensive medications. This trend is supported by recent studies, including [36], which explored the use of machine learning algorithms to predict responses to antihypertensive therapy.

This innovative approach, leveraging real-time assessments through wearable technology, streamlines monitoring, enhances treatment effectiveness, and contributes to better blood pressure control. This integration signifies a promising step towards personalized healthcare for hypertension management. In further steps, the different algorithms have to be tested on real-world data to re-evaluate its performance and robustness given real-world data.

## Appendix
## Supporting information
### Simulation

**S1 Table. Overview of systolic and diastolic blood pressure (SBP and DBP) ranges for different hypertension stages [37].**
(JPG)

**S1 Fig. Pharmacokinetic-Pharmacodynamic scheme.** top left: Relationship between time and plasma concentration given a single dose, top right: Relationship between plasma concentration and effect given a single dose, bottom: Relationship between time and effect given a single dose (based on [38]).
(TIF)

**S2 Table. Literature sources for the influence of covariates.** Based on Fig 5.
(JPG)

**S3 Table. Parameter choices for simulating the circadian rhythm, based on internal study c.f. [39].** The parameters for the average circadian rhythm are determined based on the parameter estimators provided by [39] and are further modified by the parameters derived from the average circadian rhythm observed in the internal study (see Fig 4).
(JPG)

**S4 Table. Parameter choices for simulating the medication effect using Pharmacokinetic-Pharmacodynamic (PKPD) modeling assuming normal distribution; For $E_{max}$ covariate effects need to be added (see S5 tab).**
(JPG)

**S5 Table. Parameter choices for simulating the influence of the covariates Age, Weight, Ethnicity, and Sex, on baseline blood pressure ($BSL$), maximum effect ($E_{max}$), and volume capacity ($VC$).**
(JPG)

## Modeling approaches

**Derivation of model formula for non-linear mixed effects model.** The equation can be motivated by the cumulative distribution function of the exponential distribution

$$f(x) = 1 - e^{-\lambda x}, \qquad \lambda \in \mathbb{R} \tag{20}$$

which is generalized by

$$f(x) = a - e^{-b \cdot x + c}, \qquad \lambda, a, b, c \in \mathbb{R} \tag{21}$$

so that the plateau value denoted by $a$ and the factor $b$ determining the time to reach the plateau can vary. Additionally, in our application zero must be mapped to zero since there is no medication effect prior to the first dose, which leads to

$$f(x) = a - e^{-bx + ln(a)} \quad, \tag{22}$$

which finally leads to formula (9).

## Results

**S2 Fig. Comparing predictive accuracy based on noise-free values.** Root mean squared error (RMSE) and bias across days ($t = 0 \dots, 10$) based on values without noise, models (non-linear mixed-effects model (nlme) in blue, advanced Gaussian process (GP) with five resp. ten nearest neighbours in yellow resp. red), and known measurements ($t_{max} = 1, \dots, 5$).
(TIF)

**S3 Fig. Comparison of sample size effect on root mean squared error (RMSE) and bias for observed values and noise-free values.** Given five days of measurements: (Non-linear in

yellow) mixed-effects model (nlme) in blue, advanced Gaussian process (GP) with five nearest neighbours. We assess the impact of different sample sizes on the model performance using the five nearest neighbours for the GP model. Three different scenarios are analysed, given by sample sizes of $N = 60$, $N = 120$, and $N = 200$. Again, for robustness 50 datasets are simulated and assessed for each case.

The RMSE as well as the bias are evaluated for noise-free and noisy values given 5 days of measurements. It can be seen that increasing the sample size will not change the models performance in terms of the RMSE, neither looking at the noisy nor the noise-free values. Looking at the bias, a higher sample size does not change the bias of the nlme model. Using the GP model, a higher sample size can reduce the bias.

This suggests that, given the data assumptions (such as noise, rhythm, effect size, etc.), a sample size of approximately 120 patients would be sufficient to make an adequate prediction based on 5 days of provided measurements.
(TIF)

## Acknowledgments

Katja Ickstadt acknowledges the support of BMBF and MKW.NRW within the Lamarr-Institute for Machine Learning and Artificial Intelligence. This research has been conducted using the UK Biobank Resource (Application 28807).

## Author contributions

**Conceptualization:** Berit Hunsdieck, Johanna Mielke, Katja Ickstadt, Eren Elçi.

**Formal analysis:** Berit Hunsdieck.

**Methodology:** Berit Hunsdieck, Johanna Mielke, Eren Elçi.

**Software:** Berit Hunsdieck.

**Supervision:** Katja Ickstadt.

**Visualization:** Berit Hunsdieck.

**Writing – original draft:** Berit Hunsdieck.

**Writing – review & editing:** Johanna Mielke, Katja Ickstadt, Eren Elçi.

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
