## [Decision Letter · Decision Letter 0]

8 Nov 2024

PONE-D-24-44640A Simulation-Based Framework for Modeling and Prediction of Personalized Blood Pressure Trajectories in Hypertensive Patients after Antihypertensive TreatmentPLOS ONE

Dear Dr. Hunsdieck,

Thank you for submitting your manuscript to PLOS ONE. After careful consideration, we feel that it has merit but does not fully meet PLOS ONE’s publication criteria as it currently stands. Therefore, we invite you to submit a revised version of the manuscript that addresses the points raised during the review process.

We look forward to receiving your revised manuscript.

Kind regards,

Dr. Vinod

Academic Editor

PLOS ONE

Journal Requirements:

2. Please note that PLOS ONE has specific guidelines on code sharing for submissions in which author-generated code underpins the findings in the manuscript. In these cases, all author-generated code must be made available without restrictions upon publication of the work. 

Please review our guidelines at https://journals.plos.org/plosone/s/materials-and-software-sharing#loc-sharing-code and ensure that your code is shared in a way that follows best practice and facilitates reproducibility and reuse.

4. Please note that your Data Availability Statement is currently missing the DOI/accession number of each dataset or a direct link to access each database. If your manuscript is accepted for publication, you will be asked to provide these details on a very short timeline. We therefore suggest that you provide this information now, though we will not hold up the peer review process if you are unable.

5. For studies involving third-party data, we encourage authors to share any data specific to their analyses that they can legally distribute. PLOS recognizes, however, that authors may be using third-party data they do not have the rights to share. When third-party data cannot be publicly shared, authors must provide all information necessary for interested researchers to apply to gain access to the data. (https://journals.plos.org/plosone/s/data-availability#loc-acceptable-data-access-restrictions) 

6. Please ensure that you refer to Figure 2 in your text as, if accepted, production will need this reference to link the reader to the figure.

Reviewers' comments:

Reviewer's Responses to Questions

**Comments to the Author**

1. Is the manuscript technically sound, and do the data support the conclusions?

Reviewer #1: Yes

Reviewer #2: Yes

2. Has the statistical analysis been performed appropriately and rigorously? 

Reviewer #1: Yes

Reviewer #2: Yes

3. Have the authors made all data underlying the findings in their manuscript fully available?

Reviewer #1: Yes

Reviewer #2: Yes

4. Is the manuscript presented in an intelligible fashion and written in standard English?

Reviewer #1: Yes

Reviewer #2: No

5. Review Comments to the Author

Reviewer #1: The manusript is written well including correct mathematical expressions. However, i have a few suggestions to make:

1. Abstract: first few sentences of abstract contain useless information which can be shifted to introduction. The abstract must revise the abstract completely by highlighting the significance of this work carried and some of key results obtained from this study.

2. Data in first 3 sentences of the introduction needs correction of data as per recent literature.

3. In the introduction, mention which blood pressure monitoring devices have been developed in the recent past and author must discuss the key developments in a paragraph by proper citing the literature.

4. Mention the accuracies of wrist based blood pressure monitoring devices.

5. Last paragraph of the introdcution is seems to be copied from a thesis or book. revise the introduction carefully by incorporating compititive studies.

6. Figure numbers are not in order. Check the citations of Fig. 2.

7. Figure 3 is not found in the manusript.

8. Literature described or followed are old and outdated e.g. two compartment model outlined in literature of 2004 and 2008. The simulations are fast updating everyday. Thus authors could have folllowed recent model.

Reviewer #2: The manuscript has some merits in terms of theoretical content whereas it seems to be poorly written in terms of language, which I feel must be addressed carefully by proofreading.

1. Abstract to be revised thoroughly for better clarity. Some key results of simulation study have to be presented here.

2. Keywords are not appropriate.

3. Introduction needs drastic revisions by incorporating discussion of key literature and models presented.

4. Also add a small paragraph on the key highlight of the work and novelty of the work.

5. Correct the citation of Figure 2 in the text.

6. Fig. 3 is missing.

6. PLOS authors have the option to publish the peer review history of their article (what does this mean?). If published, this will include your full peer review and any attached files.

Reviewer #1: No

Reviewer #2: No

---

## [Author Response · Author response to Decision Letter 1]

16 Jan 2025

Dear reviewers,

Thank you for your helpful comments. We apologize for the errors in the manuscript. In response to your suggestions, we have revised the mentioned parts in the updated manuscript. We carefully considered the comments and tried our best to address every one of them. We hope that these changes and the enhanced text will meet your expectations. Attached ("Response to Reviewers.pdf"), you can find the detailed response to all comments. The changes have been highlighted in green (added) and yellow (changed).

---

## [Editor Report · Decision Letter 1]

19 Jan 2025

A Simulation-Based Framework for Modeling and Prediction of Personalized Blood Pressure Trajectories in Hypertensive Patients after Antihypertensive Treatment

PONE-D-24-44640R1

Dear Dr. Hunsdieck,

We’re pleased to inform you that your manuscript has been judged scientifically suitable for publication and will be formally accepted for publication once it meets all outstanding technical requirements.

Kind regards,

Vinod Kumar Vashistha

Academic Editor

PLOS ONE
---

## [Editor Report · Acceptance letter]

PONE-D-24-44640R1

PLOS ONE

Dear Dr. Hunsdieck,

I'm pleased to inform you that your manuscript has been deemed suitable for publication in PLOS ONE. Congratulations! Your manuscript is now being handed over to our production team.

Kind regards,

on behalf of

Dr. Vinod Kumar Vashistha

Academic Editor

PLOS ONE